# Effects of Organic Elicitors on the Recycled Production of Ginkgolide B in Immobilized Cell Cultures of *Ginkgo biloba*

**DOI:** 10.3390/jfb14020095

**Published:** 2023-02-09

**Authors:** Chuang-Yu Lin, Te-Yang Huang, Wei-Chang Fu, Wen-Ta Su

**Affiliations:** 1Department of Biomedical Science and Environmental Biology, Kaohsiung Medical University, Kaohsiung 807378, Taiwan; 2Department of Orthopedic Surgery, Mackay Memorial Hospital, Taipei 104217, Taiwan; 3Department of Chemical Engineering and Biotechnology, National Taipei University of Technology, Taipei 106344, Taiwan

**Keywords:** *Ginkgo biloba*, ginkgolide B, immobilized cultivation, organic elicitors

## Abstract

*Ginkgo biloba* is a medicinal plant used in complementary and alternative medicines. *Ginkgo biloba* extracts contain many compounds with medical functions, of which the most critical is ginkgolide B (GB). The major role that GB plays is to function as an antagonist to the platelet-activating factor, which is one of the causes of thrombosis and cardiovascular diseases. Currently, GB is obtained mainly through extraction and purification from the leaves of *Ginkgo biloba*; however, the yield of GB is low. Alternatively, the immobilized cultivation of ginkgo calluses with biomaterial scaffolds and the addition of organic elicitors to activate the cell defense mechanisms were found to stimulate increases in GB production. The aim of this study was to use *Ginkgo biloba* calluses for immobilized cultures with different elicitors to find a more suitable method of ginkgolide B production via a recycling process.

## 1. Introduction

*Ginkgo biloba* is a relic plant left over from ancient times and is widely used in complementary and alternative medicines [1]. The phytochemicals in *Gingko biloba* that are considered pharmacologically active are terpenoids and flavonoids [2]. A standardized extract of *Ginkgo biloba*, EGb761, can be extracted from dried leaves as a pharmacological product. EGb761 can antagonize oxygen free radicals, stabilize cell membranes, promote neurogenesis and synaptogenesis, increase the level of brain-derived neurotrophic factors, and replicate environment-required conditions during the differentiation of stem cells into nerve cells [3,4,5]. The primary terpenoids are ginkgolides A, B, C, J, and M and bilobalide [6], which are part of the contents of EGb761. Ginkgolide B (GB) is the most noticeable ingredient with multiple pharmacological functions among the ginkgolide group [7,8,9]. The neuron-protective effects of GB have been promising on several cognitive diseases, such as Alzheimer’s and Parkinson’s diseases [8]. GB is involved in the inhibition of endothelial cell apoptosis, which can improve cardiovascular disease caused by atherosclerosis and can protect against ischemic stroke [9,10,11]. GB was found to regulate the production of nitric oxide (NO) in the body and to enhance the half-life of NO, which can extend the duration of arterial dilation and improve blood circulation. GB can also regulate NO production by modulating NOS expression and activity. The neuroprotective mechanisms of GB have been shown to enhance neuroprotective NOS activity under pathological conditions [12,13,14].

GB is a promising natural drug for the treatment of cardiovascular diseases and neuronal damage, but the content of GB in natural ginkgo trees is extremely low, at only 0.1~0.2% by weight. GB production in a callus culture of *Ginkgo biloba* was also attempted, but the yield of GB was also low, at 0.005–0.01% [15]. It is very difficult to separate and purify GB, especially at such low amounts, due to the extremely complex components in the plant tissue [16]. In 1988, the first attempt at chemically synthesizing GB was made by Corey [17], who used 1-cyclopentene and dimethoxyacetaldehyde as the substrates. After 23 intermediate reactions, GB was obtained; however, its chemical properties were not active. In 1950s, Roution and Nicell proposed a possible method of producing secondary metabolites from calluses at the industrial scale. In the 1990s, researchers successfully induced callus formation using ginkgo embryos on an MS solid medium [18]. GB was subsequently produced from callus cultures and cell suspension cultures derived from *Ginkgo biloba* leaves [19]. However, there are still many factors that hinder its industrial production in a ginkgo callus cell culture, such as the slow growth rate of the cells in vitro, the low yield of GB, its low tolerance to shear stress during cultivation, asexual mutations, and the limited yield due to the defense mechanisms of the cells. All the issues mentioned above can result in high costs for successful cultivation.

Microorganisms, plants, animal cells, or some macromolecules can attach themselves onto water-insoluble carriers via adsorption or embedding, etc., and will continue to grow, develop, reproduce, be inherited, and metabolize. A previous study showed that an immobilized culture can increase the production of terpenoid components in spite of the low production of GB [20]. This result indicated the possibility of optimizing the production of GB using an immobilized culture. The immobilized material can provide the cells with structural support so that the cells can tolerate higher shear stress, can easily form larger cell clumps, can promote cell differentiation, and can then increase the production of GB. Ginkgo cells and GB can be easily separated in an immobilized culture, and ginkgo cells can be recycled for further culture, which can save a lot of pre-cultivation time.

GB is a secondary metabolite of plants, which is a natural compound produced by plants in response to external invasion, and its production is controlled by plant defense mechanisms [21,22]. *Ginkgo biloba* cells do not synthesize large amounts of GB under normal physiological conditions. If gingko’s defense mechanism can be activated, the production of GB will be greatly increased. Some elicitors, including external environmental factors and information molecules, can activate these defense mechanism and generate the corresponding secondary metabolites by stimulating the cells through internal secretion or via additional supplementation. Elicitors that have been found include fungi, chitosan, yeast extract, and information molecules, etc. [23,24,25]. Previous studies showed that the organic elicitors methyl jasmonate (MJ) and salicylic acid (SA) can stimulate the production of secondary metabolites in cell cultures of *Saussurea medusa* and other plant species [26,27,28]; other studies also showed that MJ and SA could efficiently enhance GB production in a cell culture of *Ginkgo biloba* [29,30].

The aim of this study was to use *Ginkgo biloba* calluses for suspension and immobilized cultures to compare and contrast the differences between the two culture methods. Different elicitors were used along with a medium in order to find a more suitable method of GB production. In a suspension culture, cells suffer from shear stress, forming too small cell clusters and unstable situations, while an immobilized culture not only facilitates extracellular product recovery but also saves a lot of pre-cultivation time without the need to recover cells for reuse as a suspension culture. The addition of the elicitors helps to stimulate the defense mechanisms of ginkgo cells and increases the production of GB. Furthermore, to prevent the defense mechanism from being over stimulated, which results in damage to the ginkgo cells and GB being out of production, the recycled production of GB was attempted by introducing a recovery time between each induction for ginkgo cells. Therefore, the same batch of ginkgo cells can be cultured and can produce GB in a cultivation–induction–recovery cycle. Therefore, the combined use of an immobilized culture with proper elicitors and cycling conditions can maximize the benefits of GB production not only in terms of time consumption but also the amount of product obtained via a recycling process.

## 2. Materials and Methods

### 2.1. Ginkgo Callus Induction and Ginkgo Cell Suspension Culture Preparation and Optimal Culture Condition Analysis

To induce the ginkgo calluses, fresh young ginkgo leaves were collected and rinsed with water to remove dust. The leaves were harvested from male specimens of a ginkgo tree on the campus of Tatung University, Taipei city. The leaves were further rinsed with 70% alcohol for 30 s and, then, soaked in 2% bleach for 30 min. The bleached leaves were transferred to a sterile operation table, rinsed three times with sterilized water, cut into small pieces (2 mm^3^), and inoculated into a solid medium (1/2 MS, 3% sucrose, 0.7% agar, 1.6 g/L PVP, 0.1 g/L AA, 0.1 g/L TCA, 1 mg/L BA, 0.2 mg/L NAA, and 0.2 mg/L 2,4-D). The cut leaves were incubated in an incubator at 25 ± 1 °C and were transferred to a new solid medium every 14 days until the calluses formed. The slightly yellow and soft calluses were used for the following experiments.

Fresh ginkgo calluses (yellowish and soft) in amounts of 0.5 g, 1 g, 2 g, and 4 g were dissected individually. The dissected calluses were inoculated in 20 mL of a 1/2 MS (containing 2 mg/L of NAA, 0.5 mg/L of kinetin, 0.1 g/L of casein hydrolysate, and 0.8 g/L of PVP) medium in a 250 mL conical flask and were cultured at 30 °C and shaken with a shaker at the speed of 100 rounds per minute (rpm). The culture was incubated for 28 days, and samples were collected every 7 days for analysis. After an observation of the cell growth, the optimal culture condition was determined as follows. Two grams of ginkgo calluses was used for inoculation. The callus was inoculated in a conical flask containing 20 mL of the 1/2 MS medium in a 250 mL conical flask and was incubated at 30 °C and shaken with a shaker at the speed of 100 rpm. The calluses were cultured for 14 days and ready for subsequent treatments.

### 2.2. Selection of the Optimal Elicitors for Ginkgo Cell Suspension Culture

A series of concentrations of the elicitor solutions were prepared. Chitosan (CH) (448877-50G, Sigma-Aldrich, St. Louis, MO, USA) (50, 100, 200, 300, 400, and 500 mg/L) was dissolved in 1% acetic acid water solution. Yeast extract (YE) (Bacto™ Yeast Extract, #212750, Gibco) (50, 100, 200, 300, 400, and 500 mg/L) was dissolved in water. The methyl jasmonate (MJ) (392707-5ML, Sigma-Aldrich) (0.01, 0.05, 0.1, 0.3, and 0.5 mM) was diluted with water, and salicylic acid (SA) (S7401-500G, Sigma-Aldrich) (0.01, 0.05, 0.1, 0.3, and 0.5 mM) was dissolved in water. All of the elicitor solutions were filter sterilized before being added to the suspension cell culture individually. The cells were collected and analyzed after incubation for 2 days. The optimal conditions of the elicitors were determined in terms of cell mass, cell viability, and GB yield.

### 2.3. Analysis of Optimal Conditions of Elicitors for Ginkgo Suspension Cell Culture

The selected elicitors, MJ and SA, were added to the ginkgo suspension cell cultures to reach final concentrations of 0.5 mM and 0.3 mM, respectively. The cell mass and cell viability were analyzed after incubation with the elicitors for 2 days.

### 2.4. Effect of Different Elicitors on the Morphological Changes in Solid-State Culture of Ginkgo Cells

The cells from the suspension cell culture were removed after 14 days, and aliquots of the cells were inoculated on a 1/2 MS solid-state medium; 0.3 mM SA, 0.5 mM MJ, and 20 µL of H_2_O as a control were added to the culture. Cell morphology was observed after 2 days.

### 2.5. Selection of Suitable Immobilization Material for Ginkgo Immobilization Cell Culture

The three types of materials were obtained as follows: the sponges, typically used for aquarium filtration systems, were obtained from Amazon and had pore sizes of approximately 500 µm in diameter; the cotton filter, usually used for face masks, was obtained from Amazon and had a pore size around 500 µm in diameter; and the loofah, made from a dried loofah fiber, was purchased from a traditional market and had a pore size of 2–3 mm in diameter. Each material was cut into 1 cubic centimeter cubes as immobilized carriers (Appendix A) and added to the suspension cell culture individually. The samples were analyzed after incubation for 14 days. The cell mass, the immobilized ratio, and the structural adequacy were analyzed.

### 2.6. Selection of the Optimal Elicitors on Ginkgo Cell Immobilization Culture

After the addition of immobilized carriers into the suspension cell culture for 14 days, final concentrations of 0.3 mM SA and 0.5 mM MJ were added individually. Factors for cell defense mechanisms including PAL activity and cell viability were analyzed after incubation for 2 days.

### 2.7. Effect of Cell Recovery Time on the GB Yield Duration (Recycled Culture)

Cells under optimal culture conditions were treated with the 0.5 mM MJ elicitor for 2 days. The medium was replaced with the 1/2 MS medium without the elicitor and incubate for 0, 3, and 14 days individually. This was considered one cycle. The process was repeated for 4, 7, and 9 cycles for 14 days, 3 days, and no recovery, respectively. On the last day of each cycle, the samples were analyzed for cell viability and GB production.

### 2.8. Cell Viability Assay (MTT Activity Assay)

We performed the MTT assay by following a protocol modified from a previous study [31]. The cells were washed with PBS, detached, and resuspended in 1 mL of PBS. Twenty microliters of an MTT standard solution (50 mg/mL in PBS) was added into the cell suspension, and the cell suspension was shaken while avoiding direct exposure to light for 4 h at 37 °C. The cells were centrifuged at 2000 rpm for 10 min to remove the supernatant, 1 mL of isopropanol containing 5% (*v*/*v*) 1 M HCl, and were shaken for 4 h at room temperature. The supernatant was retrieved by centrifuging the cell suspension at 2000 rpm for 10 min, and the absorbance of A_570_ for the supernatant was measured. The MTT reading for the newly inoculated cells was set as 100% viability, and the cell viability was calculated at each designated time point.

### 2.9. Determination of Phenylalanine Ammonia Lyase Activity (PAL)

To determine the phenylalanine ammonia lyase activity, 0.1 g (FCW) of the cells was grounded with 1 mL of Tris buffer (50 mM Tris, pH8.5) in a mortar in an ice bath. The cell-crushing solution was aspirated into a microcentrifuge tube and was centrifuged at 5000 rpm for 20 min. The supernatant and reaction substrate solution (30 mM L-phenyalanine) were mixed at the ration of 1:1. The reaction was terminated by adding 20 µL of the 6N HCl solution at 37 °C for 1 h. The absorbance of A_290_ (the absorbance value of cinnamic acid) was measured. PAL activity (U) = A_290_/0.01.

### 2.10. Determination of H_2_O_2_ Concentration

To determine the H_2_O_2_ concentration, 0.1 g of the cells (FCW) was grounded with 1 mL of Tris buffer (50 mM Tris, pH 8.5) in a mortar in an ice bath. The cell-crushing solution was aspirated into a microcentrifuge tube and was centrifuged at 5000 rpm for 20 min. The absorbance value of A_240_ of the supernatant was measured. H_2_O_2_ (mM) = ∆A_240_/0.0436.

### 2.11. Catalase Activity Assay (CAT)

To conduct the catalase activity assay, 0.1 g of the cells (FCW) was grounded with 1 mL of Tris buffer (50 mM Tris, pH7.0) in an ice bath. The cell-crushing solution was aspirated into a microcentrifuge tube and was centrifuged at 5000 rpm for 20 min. In total, 50 µL of the supernatant was added into the reaction solution (Tris buffer (50 mM Tris, pH 7.0/50 mM H_2_O_2_)) at a 1:1 ration. The reaction should proceed from 15 s to 60 s at 25 °C. Changes in the A_240_ absorbance value were monitored during the proceeding reaction. H_2_O_2_ (mM) = ∆A_240_/0.0436. CAT activity (U) = 1 µmol H_2_O_2_/1 min.

### 2.12. GB Yield Analysis

For the intracellular GB assay, 0.1 g of the cells was grounded in a mortar and was dissolved by 0.5 mL of ethyl acetate. For the extracellular GB assay, 0.5 mL of the culture solution was aspirated and mixed with 0.5 mL of ethyl acetate. The extract was shaken with a shaker at 100 rpm at room temperature for 2 days and dried at 37 °C in an oven. The dried powder was dissolved with 0.5 mL methanol, and the solute was analyzed by HPLC [29]. The HPLC (YL9100 Plus, Young Lin Instrument, Anyang, Korea) used a Luna 5 μm, C18 column (4.6 × 250 mm, Phenomenex, Torrance, CA, USA), and a UV detector (Water, 2489 UV/Vis) operating at a wavelength of 250 nm. The isocratic mobile phase was a mixture of MeOH and H_2_O (50:50 (*v*/*v*)). The GB standard was purchased from Sigma-Aldrich (ginkgolide B CAS No:15291-77-7, Sigma); please see Appendix A. A GB absorption peak was detected at about 11.7 min. The GB concentration value was obtained by comparing the GB concentration and product fraction standard line.

### 2.13. Cell Morphology Observation and Starch Granule Staining by KI-I_2_

The cells were suspended in 20 mM PBS buffer (pH 7.2). A 0.5% KI-I_2_ solution was added and reacted at room temperature for 5 min. The sample was centrifuged at 1500 rpm for 2 min, and the supernatant was removed. The cells were resuspended in 20 mM PBS buffer (pH 7.2) and were observed with a light microscope.

### 2.14. Statistical Analysis

Statistical analysis was performed in Microsoft Excel 2013. Data are expressed as mean ± SD. Statistical significance was determined using a two-tailed Student’s *t*-test. An * stands for *p* < 0.05, and ** stands for *p* < 0.01.

## 3. Results

### 3.1. Gingko Cell Suspension Culture

#### 3.1.1. Determination of the Optimal Amount of Cells Required for GB Production in Suspension Culture

A 14-day culture was to be found optimal for obtaining the highest amount of ginkgo cells in terms of biomass (Figure 1a). GB is a non-growth-related secondary metabolite, and only when the cells stop growing does the GB content start to accumulate. The highest GB yield, 2 g of biomass, was the best condition for subsequent experiments (Figure 1b).

#### 3.1.2. Determination of the Optimal Cultivation Time Required for Suspension Culture and GB Production

In terms of GB production, it was observed that GB was first produced intracellularly and then slowly released extracellularly, and both of the intra- and extracellular GB contents reached 90% on day 14. According to the results of this experiment, the cell quality and GB intracellular and extracellular yields were both 90% on day 14, so 14 days was used as the standard incubation day (Figure 1c).

#### 3.1.3. Analysis for Optimal Culture Conditions of Gingko Suspension Cell Culture

Starting with 2 g of ginkgo callus seeding, after 14 days of suspension culture, the cell mass of the ginkgo cells reached 3.2 g, which was 1.6 times higher than the initial cell volume, and the cell viability remained around 95%, indicating that the cells were under good growth conditions (Figure 1d).

PAL activity, H_2_O_2_ concentration, and CAT activity were used to determine whether the plant defense mechanisms were activated. As shown in Figure 1e, the PAL activity, H_2_O_2_ concentration, and CAT activity of ginkgo cells were almost maintained at normal levels in the suspension culture without the additional of a stimulation source, with values of 26 mU/g FCW, 2.3 µM, and 9.8 U/g FCW, respectively. The results indicated that the plant defense mechanisms were not activated, which was the possible reason for the low yield of GB in the ginkgo suspension culture (Figure 1e).

### 3.2. GB Production Analysis after Addition of Elicitors

#### 3.2.1. Screening for Optimal Elicitors for GB Intracellular and Extracellular Production

Elicitors: CH (Chitosan),YE (Yeast extract),MJ (Methyl jasmonate), and SA (Salicylic acid).

The purpose of this experiment was to establish a system for recycled GB production, so we selected CH and YE as biotic elicitors, of which the structures are similar to microbial cell walls or cell components, while the organic elicitors were MJ and SA, which are two major message molecules regulating the induction of plant defense mechanisms. The results of the screenings are shown in Figure 2a–d. As the concentrations of the elicitors increase, the activation of plant defense mechanisms becomes stronger and the GB content increases significantly. However, after reaching the limit of defense mechanism activation, the GB production started to drop and formed a peak. The intra- and extracellular GB production of biological elicitors, CH and YE, reached a high point at induction concentrations of 200 mg/L and 300 mg/L, respectively, which were 47.6/69.4 mg/L and 56.4/58.4 mg/L, respectively (Figure 2a,b). Compared with the control group (without an elicitor) (23.1/22.3 mg/L), the intracellular and extracellular GB yields increased by 106%/211% for CH and 144%/162% for YE, respectively. Organic elicitors, which can activate defense mechanisms directly in plant cells, were more effective in inducing GB production, with MJ and SA reaching high GB production at induction concentrations of 0.5 mM and 0.3 mM, respectively. The GB intracellular and extracellular yields were 108.9/112.4 mg/L for MJ and 74.4/82.1 mg/L for SA, respectively (Figure 2c,d). Compared with the control group (23.1/22.3 mg/L), the yield increased by 371%/404% and 222%/268%. Based upon the result, the effectiveness of each elicitor in stimulating GB production was found to be in the following order: MJ > SA > YE > CH.

#### 3.2.2. Analysis of Optimal Elicitors for Maximizing Cell Mass and Viability of Suspension Cell Culture

Since the activation of plant defense mechanisms causes damage to the cells, in order to maximize and sustain GB production, the effects of different elicitors on the growth and viability of ginkgo cells were evaluated, as shown in Appendix A. The cell mass and viability with the biotic elicitors CH and YE reached a high point at 200 mg/L for CH and 300 mg/L for YE. The cell mass and cell viability for CH and YE were 1.83 g/71.2% and 2.13 g/81.9%, respectively, which decreased by 43.8%/25.9% and 34.6%/15.2%, respectively, compared with the cell mass and cell viability of the control group (3.26 g/97.1%). The cell mass and viability for 0.5 mM MJ and 0.3 mM SA were 2.45 g/83.4% and 2.12 g/76.4%, respectively, which decreased by 24.8%/13.7% and 34.9%/20.7%, respectively, compared with the control group. Therefore, the order of elicitors in terms of their suppression effect on cell mass and cell viability was MJ > YE > SA > CH.

Based on the results obtained above, the data were further analyzed, and the results are shown in Figure 2e. Since the yield of GB is the first criterion and the second is the viability (80 ± 5%), candidates for the subsequent experiments have to meet these requirement. MJ, which had the highest values for both GB production and viability, was selected. The second candidate selected was SA, which had the second highest GB production and viability within the acceptable range.

#### 3.2.3. Effect of Elicitors on Ginkgo-Immobilized Cell Cultures

Two grams of ginkgo calluses was cultivated for 14 days, and then, the cells were divided into six equal parts and were inoculated on the 1/2 MS solid medium. Twenty microliters of the 0.3 mM SA and 0.5 mM MJ elicitor solutions was added onto the cells, and the cells were incubated for 2 days to observe the morphological changes. The results are shown in Figure 2f. Before the addition of the elicitors, the cells appeared to be transparent and light yellow. After adding the optimal concentrations of the elicitors and incubating for 2 days, the cells without the elicitor were opaque and light yellow, while the cells with either SA or MJ turned opaque and yellow.

### 3.3. Screening of Optimal Immobilized Materials for Ginkgo-Immobilized Cell Cultures

The immobilized materials used in this experiment were a loofah, a filter cotton, and a sponge. The materials were first cut into 1 cm^3^ square cubes; then, placed in a conical flask with a working volume of 250 mL; and finally, inoculated with 2 g of ginkgo calluses. In order to select the most suitable immobilized material, the experimental results were analyzed in three aspects: cell quality, cell viability, and both intracellular and extracellular GB production. After incubating the cells for 14 days, the immobilized cells tended to form cell clusters and to differentiate, leading to growth stagnation. The immobilized cell mass was only 0.61 g in the sponge, 2.23 g in the filter cotton, and 1.13 g in the loofah, compared with 2.24 g in the suspended cells, and therefore, the cell quality was considered low (Figure 3a). However, the cell-immobilized ratios of the three materials used were 21.4%, 71.0%, and 44.0% for the sponge, filter cotton, and loofah, respectively. Based on the data obtained from the cell mass and the immobilized ratios, the structural adequacy of the materials was ranked in the following order: filter cotton > loofah > sponge.

A good immobilized cell culture system requires not only a high percentage of immobilized cells but also high cell viability. The data for cell viability are shown in Figure 3b. Among the three materials used, the viability of the immobilized cells on the sponge was significantly lower (41.2%) compared with the suspended cell fraction (64.7%). Other than that, the culture fluid for cells grown on the sponge rapidly changed colors from clear to yellow compared with those grown on either the filter cotton or loofah, which suggested that the cell defense mechanisms may be activated at different strengths by different material configurations. In order to prove our speculation, we performed a supplementary PAL activity analysis (Figure 3c) and found that, with the sponge as the immobilized substrate, the cell defense mechanism was strongly activated, which caused a significant decrease in cell viability. Although the PAL activity of either the filter cotton or loofah was increased, the activities were not significantly different from that of the control group, and their cell viability was maintained at 90% and 80% (Figure 3b), respectively. Therefore, the cell viability for the three immobilized substrates was in the following order: filter cotton > loofah fiber > sponge.

The GB yield of different immobilized materials was analyzed, as shown in Figure 3d. The GB yield of the sponge was even lower than that of the control group due to its low cell viability and immobilized cell ratio. Therefore, the sponge was first to be eliminated from being used as an immobilized material. The total GB yield of the filter cotton and loofah reached 110.8 mg/L and 146.5 mg/L, respectively, which were 154% and 236% higher than that of the control group (43.6 mg/L), and the strengths of the defense mechanism of the filter cotton and loofah were similar to that of the control group. Therefore, the GB yield results were in the following order: loofah > filter cotton > sponge.

Based on the results of the cell quality, cell viability, and GB production, the filter cotton and loofah fibers showed better performance than the sponge. In conclusion, the filter cotton and loofah fibers were selected as immobilized substrates for ginkgo cells immobilized growth and for further analysis.

### 3.4. Observation of the Callus Cells and Staining of Starch Granules

In order to demonstrate that the immobilized cell culture may contribute to cell differentiation, the change in cell morphology was recorded and the starch granules were stained with KI-I_2_, which is a stain for cytosolic starch granules. The numbers of starch granules can indicate whether the cells are proliferation-active since the starch granules usually act as energy sources for young plants. In Figure 4a, the suspended ginkgo calluses were cultured for 14 days and are mostly shown as the dark brown, round, and aggregated starch granules stained by KI-K_2_. The cells from the 14-day immobilized cultures on the cotton filter are very different from that of the suspended cell culture (Figure 4b). The more the cells differentiated, the more elongated the morphology and the less starch granules appears, which suggest that the immobilized cell culture with the filter cotton can promote cell differentiation based on the appearance of the starch granule (Figure 4b). Similar results were also observed in the immobilized cell culture with the loofah (Figure 4c).

### 3.5. Analysis for the Most Suitable Combination of the Immobilized Material and the Elicitor

We introduced the elicitors to trigger defense mechanisms in the cultured ginkgo cells. The strength of the defense mechanism is often related to the PAL activity. Thus, PAL activity was analyzed within the groups—0.3 mM SA–filter cotton, 0.3 mM SA–loofah, 0.5 mM MJ–filter cotton, and 0.5 mM MJ–loofah—and their PAL activity values were 72.0, 71, 58.5, and 54.0 mU/g FCW, respectively (Figure 5a–d), after the addition of the elicitors for 48 h. Compared with the value of the control group, 21 mU/g FCW, the PAL activity increased by 243%, 238%, 179%, and 157% for the SA–filter cotton, SA–loofah, MJ–filter cotton, and MJ–loofah groups, respectively. The results indicated that the elicitors may increase the strength of the defense mechanism by 1.5–2.5 folds compared to with the control group. Since the initiation of the defense mechanism may speed up GB production both intracellularly and extracellularly, the GB yield was estimated. Forty-eight hours after the addition of the elicitors, both the intracellular and extracellular GB productions were almost 90% greater than those of the control group. The intracellular and extracellular yields of GB for SA–filter cotton, SA–loofah, MJ–filter cotton, and MJ–loofah were 92.3/93.2 mg/L, 125.4/120.0 mg/L, 141.3/133.8 mg/L, and 165.2/172.4 mg/L, respectively, and were increased by 335%/316%, 492%/436%, 560%/497%, and 679%/670%, respectively, compared with the control group (21.2/22.4 mg/L). Among the four groups, 0.5 mM MJ–filter cotton and 0.5 mM MJ–loofah produced the second highest and the highest amounts of GB, respectively, both intracellularly and extracellularly (Figure 5c,d), while the strengths of their defense mechanisms were low among the four groups. The results may indicate that the strength of the defense mechanism and cell conditions need to reach a balance in order to keep the cells in good growth conditions while obtaining optimal GB yields. These results of GB production are summarized in Table 1.

We further tested the cell viability of the four groups: 0.3 mM SA–filter cotton, 0.3 mM SA–loofah, 0.5 mM MJ–filter cotton, and 0.5 mM MJ–loofah. Their cell viabilities were 74.5%, 69%, 81.8%, and 75% at 48h, respectively, compared with the control group (Figure 5e–h). Based on the results, the elicitor SA and the immobilized material loofah showed negative influences on the cell viability.

In spite of the cell mass produced in the loofah group being 46.3% and lower than that of MJ–filter cotton, 1.16 g of the loofah in contrast with 2.14 g of the filter cotton (Figure 3a), the difference in GB production was not significant. The data indicated that a loofah may be an ideal immobilized material for GB production. Therefore, a detailed analysis for intracellular and extracellular GB production using 0.5 mM MJ–filter cotton and 0.5 mM MJ–loofah was performed. In Figure 5i,j, the intracellular and extracellular yields of GB MJ–filter cotton and MJ–loofah were 1320/1251 µg/g FCW and 2824/2946 µg/g FCW, respectively. Compared with the GB production value of the control group, 135/140 µg/g FCW, the intracellular and extracellular yields of MJ–filter cotton and MJ–loofah increased by 8.8/7.9 folds and 19.9/20.0 folds, respectively.

### 3.6. Duration Analysis for GB Production

#### 3.6.1. Analysis of GB Production Duration (2 Days)

Recycling immobilized cell cultures for GB production is one of the reasons behind the development of this system. Therefore, under optimal conditions of the culture system, the cells were cultured for 14 days and were induced for 2 days. Thereafter, the culture medium was changed every other day and induction was continued for the next 2 days as a cycle for the GB production duration test. On the last day of every cycle, extracellular GB production and cell viability were analyzed. The results showed that GB production declined dramatically as the cycle number increased. When the ninth and seventh cycles were reached for the MJ–filter cotton and loofah groups, respectively, their GB yields were lower than that of the control group (Figure 6a). The cell viability, as expected, was also reduced as the cycle number increased. In order to improve the declining effect in the GB production duration experiment, a cell recovery phase was introduced between cycles for the defense mechanism to completely cease before the next induction.

#### 3.6.2. Impact of Time Required for Cell Recovery on GB Production Duration

It has been known that the time required for the cell defense mechanism to decline is 3 days (Figure 1e). To avoid the defense mechanism in the cells being overstimulated, the culture system was under optimal growth conditions for 14 days and induction was carried out for 2 days before it was changed to a fresh medium without elicitors for either 3 days or 14 days for cell recovery. After that, the elicitor was added for 2 days, thus completing the cycle. The results are shown in Figure 6b,c. GB production in the 3-day recovery culture showed an increasing tendency, while the GB yield in the 14-day recovery culture showed a significant increase compared with that of the cultures without recovery and with 3 days of recovery (Figure 6d). The average cell viability increased significantly in the 14-day recovery culture compared with that of the 3-day recovery culture (Figure 6e). Although the cell viability of the cultures without recovery remained high, the GB production was extremely low, on average. GB productions in the cultures with 14 days and 3 days of recovery using a cotton filter and a loofah were not significantly different (Figure 6d); however, the cell viability was better in the cultures with a cotton filter (Figure 6e). The immobilized cell culture with a recovery phase enhanced GB production and improved the cell viabilities significantly during recycled GB induction when compared with that of the suspended cell cultures.

The GB yield during the fourth cycle was 114 mg/L for the filter cotton and 102 mg/L for the loofah in the cultures with 14 days of recovery (Figure 6c). For 3 days of recovery, during the fourth cycle, the GB yield was 94 mg/L for the filter cotton and 86 mg/L for the loofah (Figure 6b). Compared with GB production during the first cycle, 157 mg/L for the filter cotton and 169 mg/L for the loofah, the values during the fourth cycle decreased by 27.4% for the filter cotton and by 39.6% for the loofah. The cell viability during the fourth cycle was 75.6% for the filter cotton and 62.1% for the loofah in the cultures with 14-day recovery (Figure 6c). For 3-day recovery, during the fourth cycle, the cell viability was 67.3% for the filter cotton and 54% for the loofah. Compared with the cell viability of the first cycle, 82.3% for the filter cotton and 76% for the loofah, the value during the fourth cycle reduced by 13.9% for the filter cotton and by 22% for the loofah (Figure 6b). We further compared the GB production and cell viability between the first cycle with recovery and without recovery. Without a recovery phase, GB production was 86 mg/L for the filter cotton and 72 mg/L for the loofah, which were 45.2% and 57.4% less, respectively, compared with that of the first cycle. For cell viability without a recovery phase, it was 63% for the filter cotton and 32% for the loofah, which decreased by 19.3% and 44%, respectively, compared with that of the first cycle. These results are summarized in Table 2.

Based on the results, the recovery time slowed down the reduction in cell viability and enhanced GB production efficiently. The decline in GB production over the cycle may be due to the defense mechanism being over-stimulated. Over-stimulation can be overcome with a different recovery time without induction, and therefore, GB production can be maintained during cycles of induction. According to the data on GB yield and cell viability, the filter cotton was found to have a lower reduction rate than that of the loofah. Therefore, we concluded that 0.5 mM MJ with a filter cotton is the optimal combination for efficient GB production.

## 4. Discussion

*Ginkgo biloba* extracts have been widely used as a healthy food supplement for a long time. The terpenoid lactones extracted from the leaves of *Ginkgo biloba*, including ginkgolides A, B, C, and J and bilobalide (BB) are functionally significant in providing anti-inflammatory, anti-atherosclerosis, anti-atherombosis, and hepatoprotective effects [32,33]. Among the effective ingredients mentioned above, also a secondary metabolite of ginkgo plants, GB is recognized as the most valuable ingredient due to its therapeutic potential and applications [7,34,35]. The amount directly extracted from ginkgo plants is fairly low, while its commercial value and needs are high. Ideally, a method of chemically synthesizing GB in a test tube would make standardizing the production process for quality control without worrying about the source of the plants easier and may be less time consuming and cost effective. However, the process is too complicated to fulfill the requirements needed for commercial scales [17,36]. Thus, seeking a practical process for GB production is needed for health improvements and for profit in business. The goal of this research was to develop an immobilized culture system from the calluses of ginkgo leaves, which can be induced to produce high concentrations of GB. In our study, we conclude that a filter cotton is the optimal immobilized material for GB production. Different immobilized materials with different pore sizes, fiber thicknesses, and structural adequacies may influence cell adhesion, growth, expansion, and GB production. The structure of the immobilized materials mimicking the natural plant environment may be the reason for the increased GB production during induction than that of the suspension culture without any structural support.

Organic elicitors have stronger effects than biotic ones on stimulating defense mechanisms, which may be due to the complete exotic features that organic elicitors bring to the cells, and the cell defense reaction is usually stronger when encountering new objects. Biotic elicitors may have certain characters that already exist in the cells, so the cell defense mechanism may not react strongly. The cells induced by MJ turned yellowish-brown, which may represent a massive accumulation of secondary metabolites in the cells (Figure 2f). The concept of using SA and MJ elicitors to activate cell defense mechanisms to enhance the production of secondary metabolites was proven to be feasible.

The results suggested that the immobilized cell culture is effective at increasing GB production and retains a high survival rate without a high-intensity defense mechanism. We also found a correlation between cell differentiation in an immobilized culture and an increase in GB yield based on the morphological change from single calluses to elongated form (Figure 4b,c). This differentiation phenomenon, instead of proliferation, seems beneficial for GB production. The intracellular chloroplasts rapidly accumulated starch granules when the cells were young and proliferation-active. A suspended cell culture produces mostly proliferated calluses, which could be the reason for the low GB yield. The cellular morphology of the immobilized cell culture is very different from that of the suspended cell culture, in which the cells are tubular and are more dispersed (Figure 4b,c). When plant cells begin to differentiate, the first organ to differentiate is the vascular tissue. The tubular cells are the primary differentiation type of vascular tissue, and therefore, the appearance of tubular cells is generally indicative of the cellular differentiation. The number of tubular cells can represent the degree of cell differentiation. The more the cells differentiate, the more elongated cells there are and more the number of starch granules decreases [37]. These results suggest that an immobilized cell culture with a filter cotton and that with a loofah can promote cell differentiation.

In the study of cell viability after induction, the elicitor SA was found to severely damage the cell viability. SA may induce and continue the strong defense mechanism in ginkgo cells, resulting in the accumulation of a large amount of secondary metabolites, which are harmful for cell growth (Figure 5e–h). The loofah has large pore sizes and thick fibers, which hinder cell adhesion and growth; its cell mass was thus relatively low compared with that of the filter cotton. The results suggested that SA and the loofah may not be suitable for GB production system development. However, we also found that the loofah, without prominent cell viability and cell mass compared with that of the filter cotton, had the highest GB yield. The results led us to include the loofah in the GB production duration analysis of its suitability along with the filter cotton.

During the GB production duration tests, the GB yield in the ninth and seventh cycles for the filter cotton and the loofah dropped below that in the control group (Figure 6a). The cell viability also decreased rapidly as the cycle number increased. This phenomenon suggests that the recycled induction of strong defense mechanisms results in severe cell damage and that GB could no longer be produced. GB is one of the secondary metabolites in ginkgo plants, and moderate production of GB can help plants overcome external disruptions. Although the source of ginkgo cells and the culture conditions are different in each study, we can still compare the incremental increase in GB production after elicitor stimulation. In a previous study, Kang et al. [29] used 1.0 mM MJ and 1.0 mM SA to increase the GB production in a suspended cell culture. The results showed eight-fold and six-fold increases in GB production for the MJ and SA treatments, respectively. However, the amount of biomass severely decreased after 48 h of treatment with the elicitors. In our study, we used 0.5 mM MJ and 0.3 mM SA to treat the suspended ginkgo cells and showed 5-fold and 3.5-fold incremental increases in GB production. Our increase ratio for GB production by elicitors is consistent with that of the previous study. In contrast, the cell viability in our study was approximately 80%, which is better than that in the previous study. In a more recent study, Sukito et al. [30] combined 0.1 mM MJ and 0.1 mM SA with jute fiber for an immobilized cell culture, and the results showed two-fold increases in GB production, while the decreased biomass in the study by Kang et al. [29] was improved using an immobilized cell culture method. In contrast, in our study, we used 0.5 mM MJ combined with a filter cotton or loofah, and the GB production increased by 5 to 7 folds (Table 1) and the cell viabilities were approximately 80%. These results indicate that combining an elicitor with immobilized materials is beneficial for GB production and ginkgo cell viability. Although these previous studies showed that the application of elicitors and immobilized culture could enhance ginkgolide B, the balance between maximizing GB production by the elicitor and maintaining ginkgo cell survival is still a critical issue for commercial/industry purposes and long-term GB production.

Our study is the first to show that an immobilized culture can be recycled for GB production for longer than 14 days. In our results, recycled cultures with a 14-day recovery showed a significance increase in GB production and cell viability compared with cultures without recovery and with 3 days of recovery, which indicated that the cells should have enough rest from elicitor stimulation and to restore the original biological conditions for further treatments. An overproduction of GB in a short period of time may conversely kill the plants themselves. Maintaining a balance between GB production and cell growth is an important issue in constructing this GB recycled production system and will be cost effective and labor-saving in future GB commercialization.

## 5. Conclusions

GB can be produced by cultured ginkgo cells in suspension or immortalized forms with organic elicitors, such as SA and MJ, which can effectively stimulate the defense mechanism of *Ginkgo*, increasing GB production by about five folds. Ginkgo-cultured cells with solid supporting materials produced at least two folds more GB than that in suspension form. Among the supporting materials used, immobilized ginkgo cells with a filter cotton is the optimal immobilized material that can effectively promote cell differentiation and increase the output of GB. Therefore, moderate stimulation of plant defense mechanisms and immobilized cells will effectively increase the production of secondary metabolites. Long lasting and strong stimulation of ginkgo-cultured cells, instead of producing more GB, could lead to death of the cells. Therefore, giving cultured cells a sufficient amount of recovery time before being recycled can increase the GB production duration. We found that a 14-day recovery time at the fourth cycle provides the best GB production condition.

## Figures and Tables

**Figure 1 jfb-14-00095-f001:**
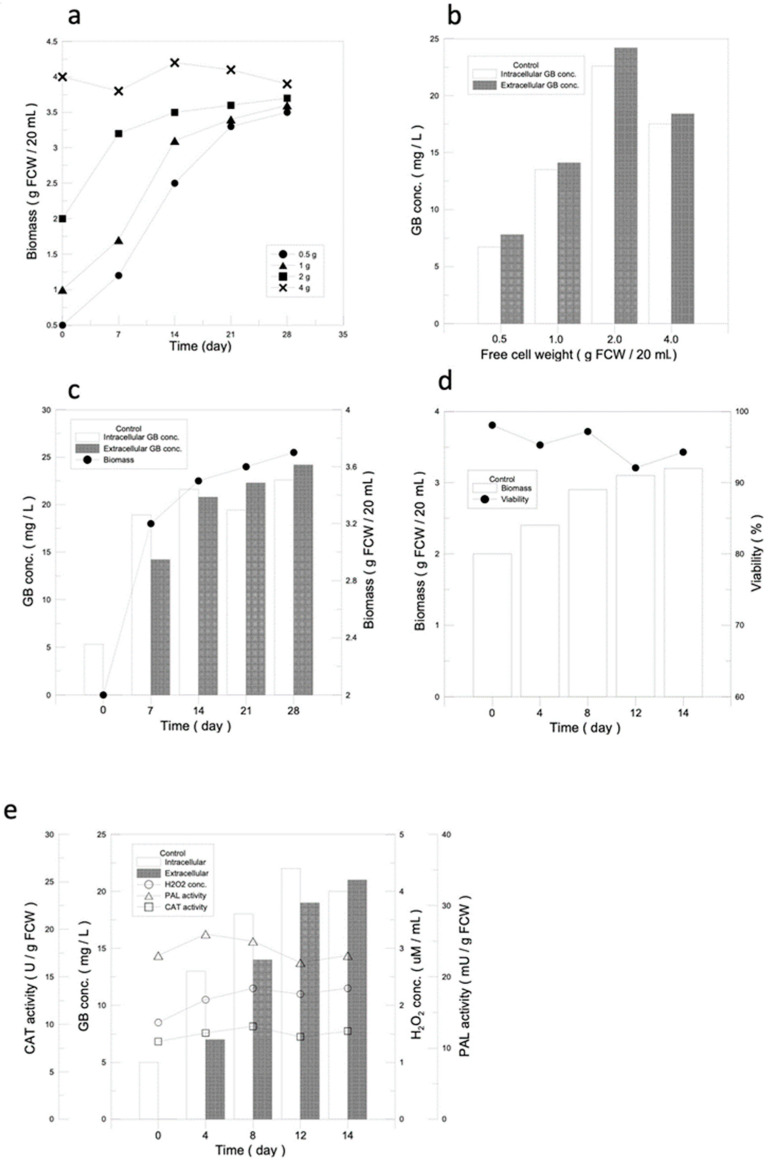
The basics behind a suspension culture of ginkgo cells. (**a**) Optimal amount of callus seeding for a ginkgo cell suspension culture. The growth curves of ginkgo suspension cells with different amounts of callus seeding. (**b**) GB produced intracellularly and extracellularly with different amounts of callus seeding. (**c**) The optimal culture time for ginkgo cell suspension culture. Relationships of intracellular and extracellular GB production and cell growth under optimal ginkgo cell suspension culture conditions. (**d**) Analysis of optimal conditions for ginkgo suspension cell culture. Curves for cell growth and cell viability under optimal conditions of ginkgo suspension cell culture. (**e**) Analysis for intracellular and extracellular GB production, PAL activity, H_2_O_2_ concentration, and CAT activity under optimal conditions of ginkgo suspension cell culture.

**Figure 2 jfb-14-00095-f002:**
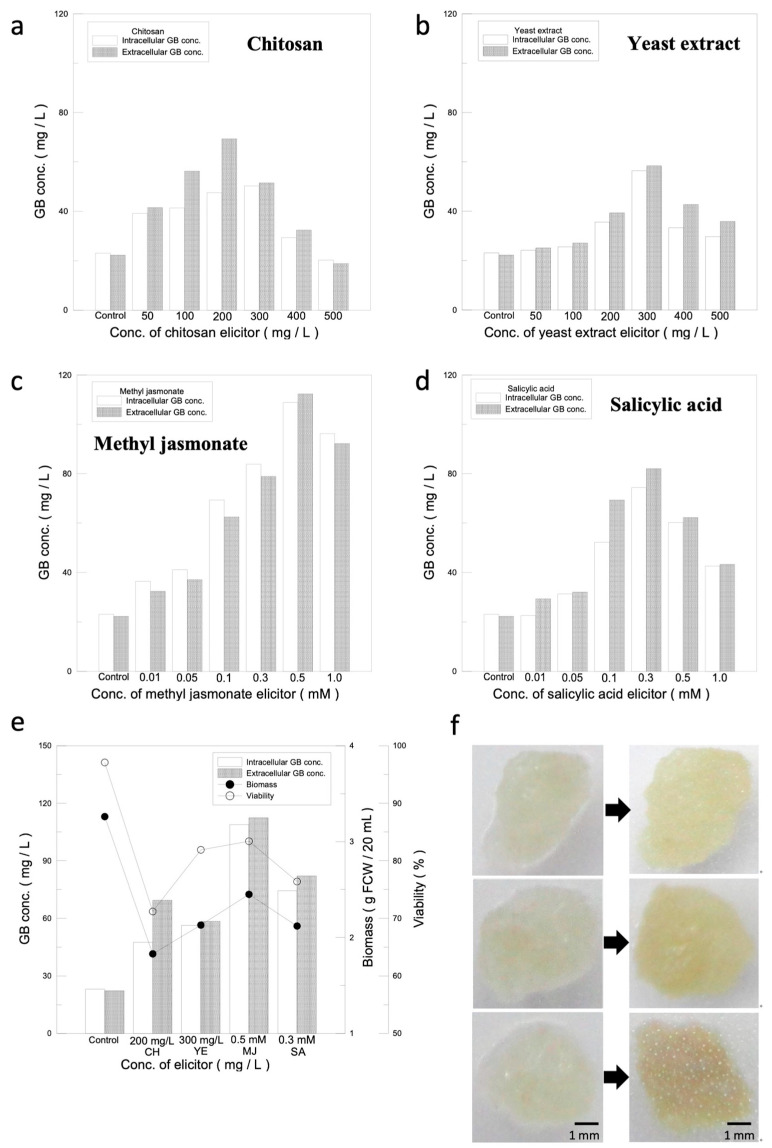
Elicitor stimulation analysis. (**a**–**d**) Selection of the optimal elicitor for ginkgo cell suspension culture based on intracellular and extracellular GB production. Impact on GB production of different concentrations of elicitors. (**e**) Analysis of optimal concentrations for elicitors. Relationship among GB production, cell mass, and cell viability under optimal concentrations of each elicitor. (**f**) Impact of different elicitors on ginkgo-immobilized cell culture. Impact of SA and MJ elicitors on morphology of ginkgo-immobilized cell culture. The left panel: day 0, right panel: day 2. From top to bottom: without elicitor, 0.3 mM SA and 0.5 mM MJ.

**Figure 3 jfb-14-00095-f003:**
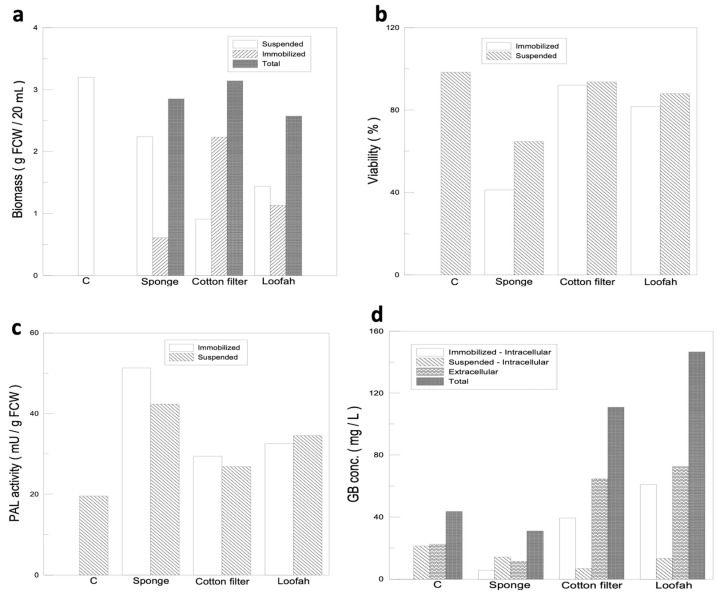
Immobilized cell culture and selection for optimal immobilized material for ginkgo-immobilized cell culture. (**a**) Comparisons of ginkgo cell mass using different immobilized materials. (**b**) Comparisons of ginkgo cell viability using different immobilized materials**.** (**c**) PAL activity of ginkgo cells growing on different immobilized materials**.** (**d**) Comparisons of intracellular and extracellular GB production using different immobilized materials.

**Figure 4 jfb-14-00095-f004:**
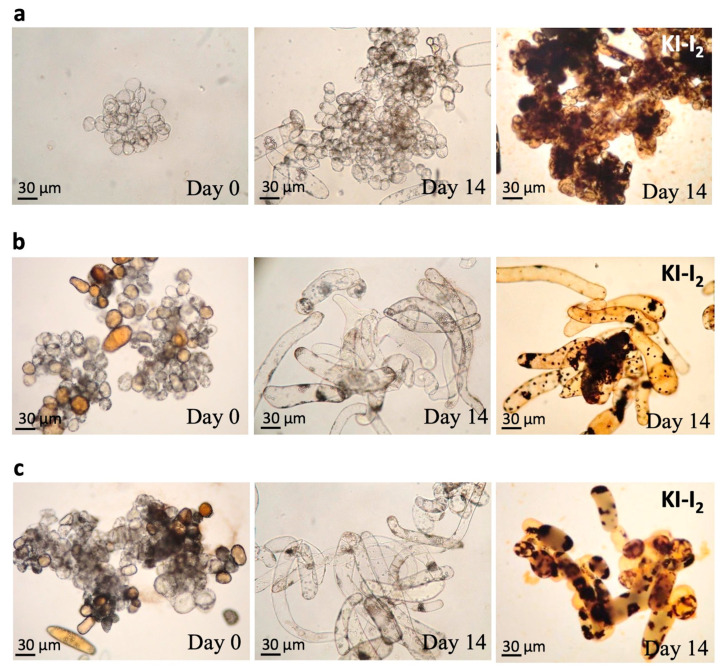
Morphological change and starch granules (dark brown) in calluses cultured on cotton filter and loofah on day 0 and day 14. (**a**) Suspended callus cells. (**b**) The immobilized calluses on cotton filter. (**c**) The immobilized calluses on loofah.

**Figure 5 jfb-14-00095-f005:**
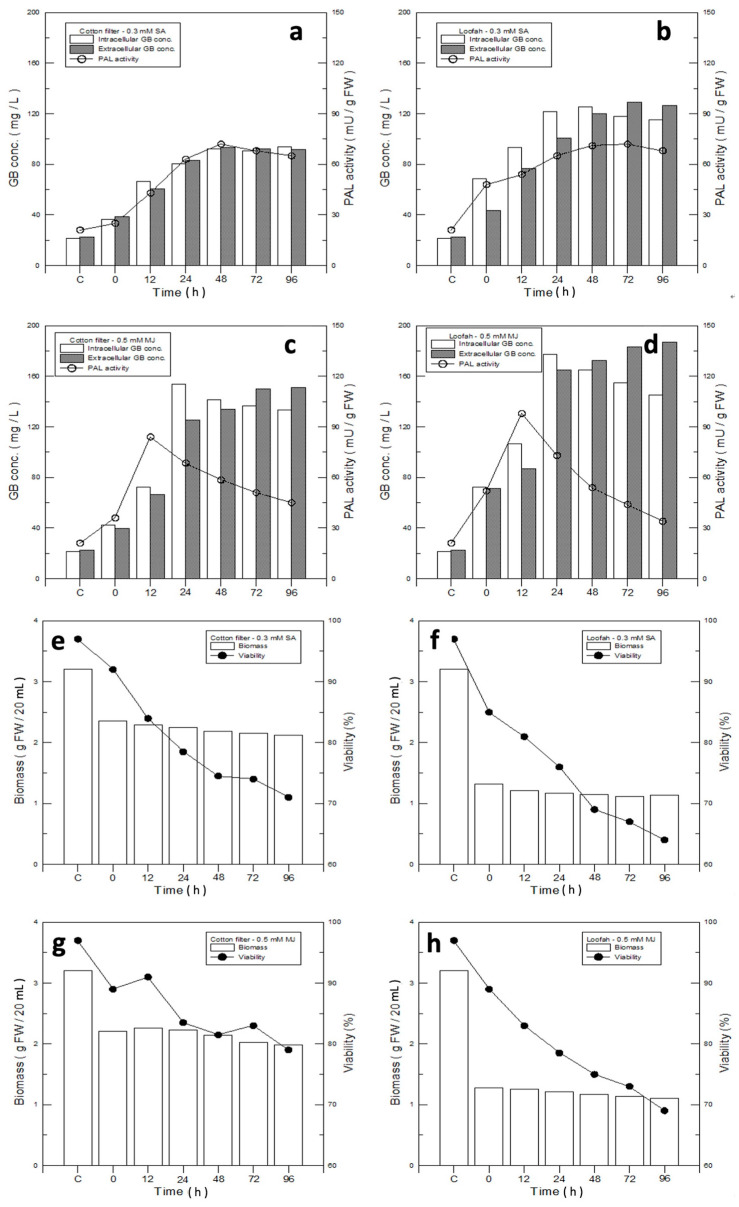
Analysis of optimal immobilized materials and elicitors for ginkgo-immobilized cell culture. (**a**–**d**) Relationships of GB production and PAL activity with different elicitors and immobilized materials: (**a**) 0.3 mM SA–filter cotton; (**b**) 0.3 mM SA–loofah; (**c**) 0.5 mM MJ–filter cotton; and (**d**) 0.5 mM MJ–loofah. (**e**–**h**) Comparison of biomass and cell viability of ginkgo cell-immobilized culture on filter cotton and loofah with SA or MJ elicitors: (**e**) 0.3 mM SA–filter cotton; (**f**) 0.3 mM SA–loofah; (**g**) 0.5 mM MJ–filter cotton; and (**h**) 0.5 mM MJ–filter cotton. (**i,j**) GB concentration and PAL activity of ginkgo cells growing on (**i**) 0.5 mM MJ–filter cotton and (**j**) 0.5 mM MJ–loofah.

**Figure 6 jfb-14-00095-f006:**
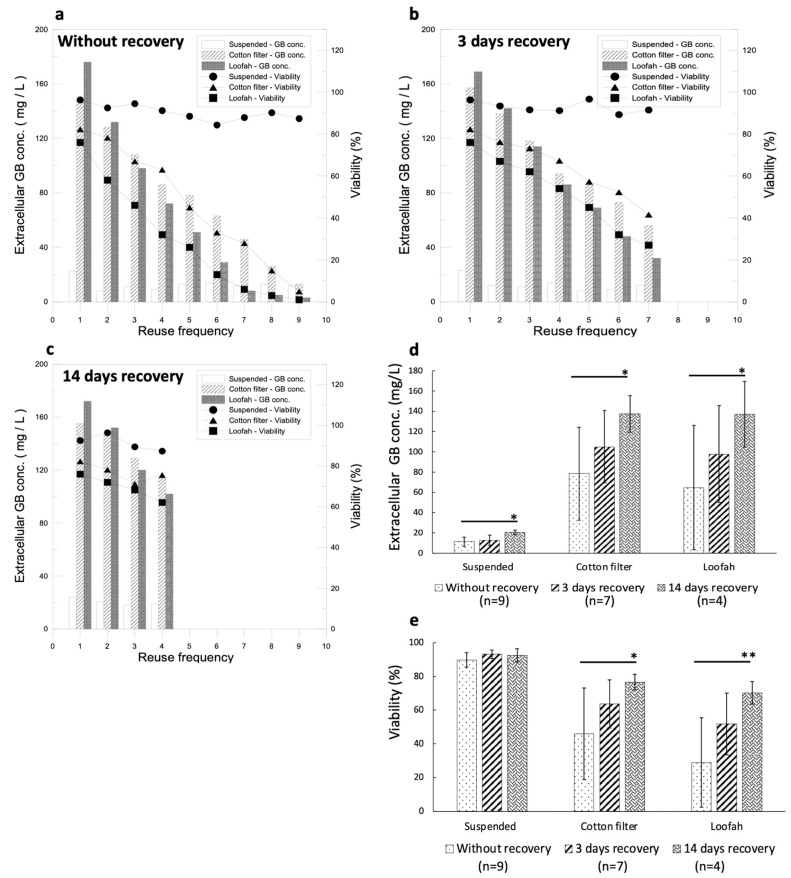
Optimization of the recycled immobilized cell culture for GB production conditions for the culture system. (**a**) Analysis of GB production duration under optimal conditions of cell culture and elicitors (cycling conditions: induction for 2 days without recovery). (**b**) Analysis of GB production duration under optimal immobilized culture and optimal elicitor conditions (cycling condition: induction for 2 days with 3-day recovery between inductions). (**c**) Analysis of GB production duration under optimal immobilized culture and optimal elicitor conditions (cycling condition: induction for 2 days with 14-day recovery between inductions). (**d**) The average GB production with different recovery days. (**e**) The average cell viabilities with different recovery days. (An * stands for *p* < 0.05 and ** stands for *p* < 0.01).

**Table 1 jfb-14-00095-t001:** GB production in different treatments (mg/L).

w/o Elicitors	Suspended Cells	Sponge	Filter Cotton	Loofah	w/o Immobilized Materials
**Total GB production**	ND	<43.6	110.8	146.5	43.6

**Suspended cells with elicitors**	CH	YE	MJ	SA	w/o elicitors
Intracellular GB production	47.6	56.4	108.9	74.4	23.1
Extracellular GB production	69.4	58.4	112.4	82.1	22.3

**Immobilized cells with elicitors**	SA/filter cotton	SA loofah	MJ/filter cotton	MJ/loofah	Suspended cells w/o elicitors
Intracellular GB production	92.3	125.4	141.3	165.2	21.2
Extracellular GB production	93.2	120	133.8	172.4	22.4

**Table 2 jfb-14-00095-t002:** The cell viability and GB yield of cycle 4 for 14-day and 3-day recovery, and those of cycle 1 and without recovery using the filter cotton and loofah as immobilized materials.

**Cycle 4—14 Day Recovery**		**Filter Cotton**	**Loofah**
	Viability	75.6%	62.1%
	GB yield	114 mg/L	102 mg/L

**Cycle 4—3 Day Recovery**		**Filter Cotton**	**Loofah**
	Viability	67.3%	54%
	GB yield	94 mg/L	86 mg/L

**Cycle 1**		**Filter Cotton**	**Loofah**
	Viability	82.3%	76%
	GB yield	157 mg/L	169 mg/L

**No Recovery**		**Filter Cotton**	**Loofah**
	Viability	63%	32%
	GB yield	86 mg/L	72 mg/L

## Data Availability

All relevant data are reported in the article.

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
