# Peer review of "Effects of Organic Elicitors on the Recycled Production of Ginkgolide B in Immobilized Cell Cultures of Ginkgo biloba"

_jfb, 2023, doi:10.3390/jfb14020095_

Round 1

Reviewer 1 Report (New Reviewer)

The paper can be published in JFB after the major revision. The authors examined the production of ginkgolide B in cell cultures of Ginkgo biloba. The topic fits in with the journal's scope. The authors investigated the effect of different materials used for cell immobilization and the addition of various biotic and abiotic elicitors. The research is very extensive, which probably caused difficulties in describing the results precisely and clearly. The literature items have all been used in the text. However, the authors used only 28 references. With a very rich bibliography on Ginkgo biloba, I consider it insufficient. I appreciate the work the authors put into this paper. However, this paper needs a lot of corrections:

Introduction:

For chitosan, yeast extract, methyl jasmonate and salicylic acid, the correct term is elicitors, not inducers. The same applies to the rest of the text. Please give examples of the use of elicitors in Ginkgo biloba cultures.

Materials and Methods:

1.     Please fill in where the leaves used to initiate the in vitro cultures came from and whether they were male or female specimens.

2.     "dissect and remove..." the sentence is incomprehensible

3.     "100 rpm" what apparatus was used?

4.     How were the elicitor solutions prepared, what was the solvent, how was the sterility of the added solutions obtained? What were the controls, were solvent alone added to them?

5.     What does the abbreviation “RT” stand for?

6.     Why are there no literature references in the methodology of the MTT and other tests?

7.     The description of HPLC analyzes lacks information about the apparatus used, detector, etc. The source of the ginkgolide B standard is not specified. Please explain why the detection was carried out at a wavelength of 250 nm when it is known that this compound has a different absorption maximum? I am asking the authors to include chromatograms of the extracts and the standard substance as well as UV spectra of the compounds in the extracts and the standard in the additional materials in order to authenticate the identification.

Results:

  1. Please improve the resolution of the figures. They are illegible in their present form
  2. I would like to ask the authors to be consistent in the terms used regarding in vitro cultures. Since cell cultures were obtained, the term ”tissue” cannot be used (description of Figure 1); Does the term "seeding" mean inoculum?
  3. The description of the results is unclear to the reader. A large amount of information does not allow reading the most important results. Please collect all ginkgolide B results in one table, add the standard deviation, specify how many repetitions there were for each sample. Then it will be easier for the reader to see which method gives the best results.
  4. Very important - I would like to ask for statistical analysis of the results.
  5. At the end of the Results section, the authors describe the results for cycles 4 and 1. The materials and methods do not explain what these terms mean and how exactly they were followed. Thus, the discussed results are completely incomprehensible and it is not known how to relate them to the previously discussed results.

Discussion:

  1. References are in the wrong brackets
  2. The discussion lacks a comparison of own results with those of other authors.

Conclusion:

  1. Please carefully list the most important results, which of them are the most promising, which may have practical application.
  2. Why did the authors use the term immobilized callus and not immobilized cells? This is confusing in the context of previously reported results that suggested immobilization of cells obtained from a cell suspension. The callus tissue was used to initiate the cell suspension, I understand.

References:

Please correct the literature as required by the journal.

An additional note:

Please correct the article in editorial and linguistic terms. I noticed typos and stylistic errors.

Author Response

Reviewer 1:

The paper can be published in JFB after the major revision. The authors examined the production of ginkgolide B in cell cultures of Ginkgo biloba. The topic fits in with the journal's scope. The authors investigated the effect of different materials used for cell immobilization and the addition of various biotic and abiotic elicitors. The research is very extensive, which probably caused difficulties in describing the results precisely and clearly. The literature items have all been used in the text. However, the authors used only 28 references. With a very rich bibliography on Ginkgo biloba, I consider it insufficient. I appreciate the work the authors put into this paper. However, this paper needs a lot of corrections:

We appreciate the reviewer’s critical comments. The manuscript has modified according to the reviewer’s suggestions. The literature items are increased and modified the presentation of results and conclusions. Please refer to the modified manuscript.

Introduction:

For chitosan, yeast extract, methyl jasmonate and salicylic acid, the correct term is elicitors, not inducers. The same applies to the rest of the text. Please give examples of the use of elicitors in Ginkgo biloba cultures.

We agree with the reviewer’s comment and correct the term of inducers to elicitors in the manuscript. The introduction part also modified according to the reviewer’s suggestion. Please refer to the modified manuscript.

Materials and Methods:

  1. Please fill in where the leaves used to initiate the in vitro cultures came from and whether they were male or female specimens.

According to the reviewer’s comment, we modified the materials and methods part in 2.1. please refer to the modified manuscript.

  1. "dissect and remove..." the sentence is incomprehensible

According to the reviewer’s comment, we modified the manuscript accordingly.

  1. "100 rpm" what apparatus was used?

According to the reviewer’s comment, we modified the manuscript. It stands for the Centrifuge speed in 100 round-per-minute (rpm).

  1. How were the elicitor solutions prepared, what was the solvent, how was the sterility of the added solutions obtained? What were the controls, were solvent alone added to them?

We have modified and described the details in Materials and Methods of the manuscript according to reviewer’s comment.

  1. What does the abbreviation “RT” stand for?

According to the reviewer’s comment, we modified the manuscript. The “RT” stands for the room temperature.

  1. Why are there no literature references in the methodology of the MTT and other tests?

According to the reviewer’s comment, we modified the manuscript and add a reference accordingly.

  1. The description of HPLC analyzes lacks information about the apparatus used, detector, etc. The source of the ginkgolide B standard is not specified. Please explain why the detection was carried out at a wavelength of 250 nm when it is known that this compound has a different absorption maximum? I am asking the authors to include chromatograms of the extracts and the standard substance as well as UV spectra of the compounds in the extracts and the standard in the additional materials in order to authenticate the identification.

We have modified the description of HPLC on Materials and Methods part. Please refer to the manuscript. The chromatograms are also provided in supplementary materials S3.

Results:

  1. Please improve the resolution of the figures. They are illegible in their present form

According to reviewer’s comment, we modified the figures to higher image quality. Please refer to the manuscript.

  1. I would like to ask the authors to be consistent in the terms used regarding in vitro cultures. Since cell cultures were obtained, the term ”tissue” cannot be used (description of Figure 1); Does the term "seeding" mean inoculum?

According to the reviewer’s comment, we have corrected the term to “cells”. Please refer to the manuscript. The term “seeding” stands for inoculation.

  1. The description of the results is unclear to the reader. A large amount of information does not allow reading the most important results. Please collect all ginkgolide B results in one table, add the standard deviation, specify how many repetitions there were for each sample. Then it will be easier for the reader to see which method gives the best results.

According to the reviewer’s comment, we have modified the results and make the Table 1 to illustrate the results of GB. About statistical analysis, please refer to the statements in next reply.

  1. Very important - I would like to ask for statistical analysis of the results.

We appreciate the reviewer’s critical comments and suggestions. Here We would like to illustrate a qualitative approach for this study and proof a concept of cycled production of GB. Because of some reasons, eq. the shortage of experimental materials, the laboratory labor power, etc., it’s difficult to collect enough results for statistical analysis. These experiments were well conducted and carefully verified in a reliable way under very limited conditions. Therefore, we hope the reviewer would understand our limitations and we will improve the statistical analysis in the future as long as the supply is sufficient.

  1. At the end of the Results section, the authors describe the results for cycles 4 and 1. The materials and methods do not explain what these terms mean and how exactly they were followed. Thus, the discussed results are completely incomprehensible and it is not known how to relate them to the previously discussed results.

According to the reviewer’s comment, we have modified the manuscript. Please refer to the modified manuscript.

Discussion:

  1. References are in the wrong brackets

According to reviewer’s comment, we have modified the format of references. Please refer to the modified manuscript.

  1. The discussion lacks a comparison of own results with those of other authors.

According to reviewer’s comment, we modified the manuscript. Please refer to the modified manuscript.

Conclusion:

  1. Please carefully list the most important results, which of them are the most promising, which may have practical application.

According to the reviewer’s comment, we have modified the conclusion part. Please refer to the modified manuscript.

  1. Why did the authors use the term immobilized callus and not immobilized cells? This is confusing in the context of previously reported results that suggested immobilization of cells obtained from a cell suspension. The callus tissue was used to initiate the cell suspension, I understand.

According to the reviewer’s comment, we have modified the term to immobilized cells for consistency. Please refer to the modified manuscript.

References:

Please correct the literature as required by the journal.

According to reviewer’s comment, we modified the format of literatures. Please refer to the modified manuscript.

An additional note:

Please correct the article in editorial and linguistic terms. I noticed typos and stylistic errors.

According to the reviewer’s comment, we have corrected the terms following the journal’s requirements. Please refer to the modified manuscript.

Reviewer 2 Report (Previous Reviewer 2)

It seems that the authors respected the previous suggestions made by the reviewers.

However, the conclusion section should briefly highlight the main results obtain in the study and further present the conclusions to provide a logical conclusion of the study.

Author Response

Reviewer 2:

It seems that the authors respected the previous suggestions made by the reviewers.

However, the conclusion section should briefly highlight the main results obtain in the study and further present the conclusions to provide a logical conclusion of the study.

We appreciate the reviewer’s comment and we have modified the conclusion section of manuscript according to reviewer’s suggestion. Please refer to the modified manuscript.

Reviewer 3 Report (Previous Reviewer 4)

The Authors have improved the manuscript in many points, have added cell morphology observation and starch granules staining by KI, have added a photograph of materials used for immobilisation. However, the main problem (a lack of statistical analysis) is without any changing. The Authors wrote, that they have not been able to made statistical analysis because of several limitations. I understand, but I wonder how many replications they have made for the experiments? They have not matched the statistical errors or deviations on the graphs. There is also lack of such information in the material and method section.  All these points suggest that the Authors have not made enough repetitions to draw conclusions. Therefore, in my opinion the paper has severe flaws, but I left the final decission to the Editor. 

Author Response

Reviewer 3:

The Authors have improved the manuscript in many points, have added cell morphology observation and starch granules staining by KI, have added a photograph of materials used for immobilisation. However, the main problem (a lack of statistical analysis) is without any changing. The Authors wrote, that they have not been able to made statistical analysis because of several limitations. I understand, but I wonder how many replications they have made for the experiments? They have not matched the statistical errors or deviations on the graphs. There is also lack of such information in the material and method section. All these points suggest that the Authors have not made enough repetitions to draw conclusions. Therefore, in my opinion the paper has severe flaws, but I left the final decission to the Editor.

We appreciate the reviewer’s comments and suggestions. Because we encountered the shortage of experimental materials and the laboratory labor power problem, it’s difficult to collect enough results for statistical analysis. These experiments were well conducted and carefully verified in a reliable way. We are trying to deliver the concept in developing the recycled GB production platform with qualitative approaches in this study under very limited conditions. Therefore, we hope the reviewer would understand our limitations and we will improve the statistical analysis in the future as long as the supply is sufficient.

Round 2

Reviewer 1 Report (New Reviewer)

The authors corrected the text according to the comments and answered my questions. The article may be published.

Author Response

Reviewer 1

  1. The authors corrected the text according to the comments and answered my questions. The article may be published.

We appreciate the reviewer’s comments and suggestions to optimize our manuscript.

Reviewer 3 Report (Previous Reviewer 4)

The Authors have improved the paper in many points. But they have not been able to make statistical analysis because of few repetitions of the experiments (which I have understood, but it is hard to draw proper conclusions if the number of repetitions is not sufficient). In one experiment I have found information that the experiments were repeated 4, 7 and 9 times. It is sufficient for statistical analysis as well as for calculations of standard deviation (error), which are absolutely minimal requirements, but I can not find SD nor SE in any figure. 

Author Response

Reviewer 3

  1. The Authors have improved the paper in many points. But they have not been able to make statistical analysis because of few repetitions of the experiments (which I have understood, but it is hard to draw proper conclusions if the number of repetitions is not sufficient). In one experiment I have found information that the experiments were repeated 4, 7 and 9 times. It is sufficient for statistical analysis as well as for calculations of standard deviation (error), which are absolutely minimal requirements, but I can not find SD nor SE in any figure.

We appreciate the reviewer’s comments and suggestions. According to the reviewer’s comment, we have added Figure 6-d and e to show the results of statistical analysis. Please refer to the modified manuscript.

This manuscript is a resubmission of an earlier submission. The following is a list of the peer review reports and author responses from that submission.

Round 1

Reviewer 1 Report

The manuscript was well written and the findings have a significant level of novelty. However, the resolution of the figures both in the manuscript and the supplementary file must be improved as they are almost blunt and unreadable.

The findings should also be discussed further and more articles should be referred to, as there are many recently published articles on the area including "Therapeutic promises of ginkgolide A: A literature-based review by Sarkar et al. 2020 and many more that could make the discussion richer. 

Some of the references are too old to refer to in the current trend of research and perhaps they should be replaced with the most recent ones.

Author Response

Reviewer1

Comments and Suggestions for Authors

The manuscript was well written and the findings have a significant level of novelty. However, the resolution of the figures both in the manuscript and the supplementary file must be improved as they are almost blunt and unreadable.

The findings should also be discussed further and more articles should be referred to, as there are many recently published articles on the area including "Therapeutic promises of ginkgolide A: A literature-based review by Sarkar et al. 2020 and many more that could make the discussion richer. 

Some of the references are too old to refer to in the current trend of research and perhaps they should be replaced with the most recent ones.

We appreciate reviewer’s comments. We have changed the figures to better resolution images. The references are updated according to reviewer’s suggestion. The discussion parts are also revised. Please refer to the revised article.

Author Response

Reviewer2

The manuscript regarding the Effects of Abiotic Inducers on the Continuous Production of Ginkgolide B by Immobilized Cell Cultures of Ginkgo biloba is overall a good original research. However, I recommend some minor adjustments: The quality of the images must be improved. Also, the discussion section should include more data by comparing the present research with other similar articles related to immobilized plant cell cultures – what is similar and what is different in their approach. The conclusion section should me more elaborated.

We appreciate reviewer’s comments. We have revised the article according to reviewer’s suggestion. Please refer to the revised article.

Reviewer 3 Report

Ginkgo biloba should be in script font throughout the manuscript.

All figures should be presented in high resolution. Some of these figures are difficult to visualize.

Grammar and spelling should be checked throughout the manuscript.

Methods. The methodology of the quantitation of ginkgolide B should be described in detail. The analytical conditions should be indicated.

How were selected the concentrations of inducers?

Discussion. The results (% yield) should be compared with other methodologies of ginkgolide B production. The advantages and disadvantages of the current protocol should be discussed in comparison with other methodologies.

Author Response

Reviewer3

Comments and Suggestions for Authors

Ginkgo biloba should be in script font throughout the manuscript.

 All figures should be presented in high resolution. Some of these figures are difficult to visualize.

Grammar and spelling should be checked throughout the manuscript.

 Methods. The methodology of the quantitation of ginkgolide B should be described in detail. The analytical conditions should be indicated.

 How were selected the concentrations of inducers?

 Discussion. The results (% yield) should be compared with other methodologies of ginkgolide B production. The advantages and disadvantages of the current protocol should be discussed in comparison with other methodologies.

We appreciate reviewer’s comments. The figures are changed to better resolution images. The references are updated according to reviewer’s suggestion. The discussion parts are also revised. Please refer to the modified article.   

Author Response

Reviewer4

The reviewed paper is quite interesting, mainly because of importance of the metabolites obtained from the plant. However, I have some notes and questions, which have been listed below:

  1. How was the callus tissue of Gingko obtained? There is no information about the origin and age (numer of subcultures, time in in vitro culture) of the described callus culture. We described the details in materials and methods
  2. What was the novelty of the paper? It should be highlighted more, especially in. the aim of the study. The novelty of this article is described in the introduction part.
  3. l. 106 Is it a whole sentence? The sentence was redundant and was omitted.
  4. Fig. 2f. There is lack of scale bar in the picture. Moreover, the photo is not good quality (on my computer screen). The images are changed to better resolution ones and the scale bar has been added.
  5. In my opinion, for better comparison, statistical significance analyses are absolutely necessary for results presented in Fig. 1-5. We agree with reviewer’s suggestion and will perform it in the future works.
  6. In my opinion photos of materials used for immobilisation (before and/or after immobilisation) should be added. We agree with the reviewer’s suggestion and will perform it in the future works.
  7. All latin names in the whole paper should be written with italyc font.

We modified it according to the reviewer’s suggestion.

  1. Which unit is in fig. 5 on axis X (reuse frequency)? Yes, it is frequency.
  2. In my opinion, discussion section is poorly written in terms of previous papers on suspension cultures of Gingko sp., incluidng elicitation and production of metabolites in the cultures. We modified the discussion part according to reviewer’s suggestion.
  3. Why the Authors have chosen the materials for immobilisation? Any literature. data, preliminary studies? We have chosen these materials according to some of our previous studies.
  4. Because of plenty results described in the paper, a reader is little confused. The Authors tried to summarize the most important results, but in my opinion they can do it better. It is not clear which material for immobilisation seems to be the best and why? To present the results more clearly, we added a table (Table 1) for summary.

To conclude, I recommend reconsider the paper for publication but after major revision.

We appreciate the reviewer’s comments. The figures are changed to better resolution images. The references are updated according to reviewer’s suggestion. The discussion parts are also revised. Please refer to the modified article.

Round 2

Reviewer 3 Report

The authors did not attend to my recommendations for improving the manuscript.

Author Response

Reviewer3

Comments and Suggestions for Authors

  1. Ginkgo biloba should be in script font throughout the manuscript.

We have modified the manuscript according to reviewer’s comment.

  1. All figures should be presented in high resolution. Some of these figures are difficult to visualize.

According to reviewer’s comment, all the figures have updated to higher resolution and quality.

  1. Grammar and spelling should be checked throughout the manuscript.

We have checked and corrected grammar and spelling in the manuscript.

  1. Methods. The methodology of the quantitation of ginkgolide B should be described in detail. The analytical conditions should be indicated.

According to reviewer’s suggestion, we described the method of GB analysis and quantification in page 4, 2.12. GB yield analysis.

  1. How were selected the concentrations of inducers?

According to our previous experimental tests, the concentrations of the inducers were determined.

  1. Discussion. The results (% yield) should be compared with other methodologies of ginkgolide B production. The advantages and disadvantages of the current protocol should be discussed in comparison with other methodologies.

We appreciate the reviewer’s comment. The ways to present the GB yield are varied in other studies, eq. mg/g (dry weight), mg/L (medium), mg/kg, etc., and cannot be standardized from the system to system. It is difficult for us to compare the effectiveness of GB production without standardizing the units for GB yield. The purity of GB is another concerned issue in comparing the results with other studies. To better address this issue, we have cited references #24, 25, 26 and 27 to address the results from other studies in the discussion.

Reviewer 4 Report

I appreciate the effort of Authors. An improvement of reviewed paper is visible. The Authors have corrected the paper according to my almost all points. However, I have additional question for a statement of the Authors that they will make statistical analyses in their future work. I can not understand if the future work will contain the same results or different? Statistical analyses are crucial for comparison of the significance of obtained results and for comparison of means between variants in the present, described in the paper results. The same is with the photos of materials used in the present paper. A reader will be able to imagine the materials if they will be presented on a photo.

Author Response

Reviewer 4

I appreciate the effort of Authors. An improvement of reviewed paper is visible. The Authors have corrected the paper according to my almost all points. However, I have additional question for a statement of the Authors that they will make statistical analyses in their future work. I can not understand if the future work will contain the same results or different? Statistical analyses are crucial for comparison of the significance of obtained results and for comparison of means between variants in the present, described in the paper results. The same is with the photos of materials used in the present paper. A reader will be able to imagine the materials if they will be presented on a photo.

We appreciate reviewer’s critical comments and suggestions. Because of some reasons, eq. the shortage of experimental materials, the laboratory labor power, etc, it’s difficult to collect enough results for statistical analysis. These experiments were well conducted and carefully verified in a reliable way. We are trying to deliver the concept in developing the continuous GB production platform in this study under very limited conditions. Therefore, we hope the reviewer would understand our limitations and we will improve the statistical analysis in the future as long as the supply is sufficient.

The photos of the immobilized materials are shown in the supplementary part. 

Round 3

Reviewer 3 Report

The manuscript can be accepted